# A Narrative Literature Review Using Placemaking Theories to Unravel Student Social Connectedness in Hybrid University Learning Environments

**Theresa Wheele [1,2,*]** , **Clara Weber [2,3,*]**, **Lukas Windlinger [1]** , **Tore Haugen [2]** and **Carmel Lindkvist [2,*]**

1 Workplace Management, Institute of Facility Management, Zürich University of Applied Sciences, 8820 Zurich, Switzerland

2 Institute of Architecture and Planning, Norwegian University of Science and Technology, 7491 Trondheim, Norway

3 School of Psychology, Faculty of Health and Medical Sciences, University of Surrey, Guildford GU3 7XH, UK

\* Correspondence: theresa.wheele@zhaw.ch (T.W.); clara.weber@zhaw.ch (C.W.); carmel.lindkvist@ntnu.no (C.L.)

**Abstract:** Student social connectedness is indicated to be changing with the increasing digitalisation of universities. This narrative literature review aims to bring new meanings to the hybrid university learning environment (HULE), and to develop a framework for the benefit of student social connectedness by using placemaking theories. It searches through the academic literature for evidence of experience with three attributes of social connectedness: socialising, social support, and sense of belonging, in relation to individuals' sense of place, bringing a range of outcomes, such as identity development, which might contribute to improved social connectedness. This is then expanded in the HULE by looking at the physical and online spaces, with a focus on liminal space and co-design. The findings show that an overly rigid structure of the HULE can cause negative student social connectedness, with co-design being proposed as a way of creating a tailored and connected learning experience. However, this is underdeveloped for learning environment needs and could be enhanced by applying placemaking theories to map levels of student social connectedness in the physical location and in the more-than-real 'non-places'. This provides an innovative perspective of the HULE based on student social connectedness, impacting the existing pedagogical approach for university courses.

**Keywords:** social connectedness; hybrid; student experience; learning environment; placemaking

## 1. Overview

### 1.1. Background

University environments are being propelled to change after restrictions of the COVID-19 pandemic and developments in digitalisation [1–3] over the last decade. These changes have brought a growing movement away from traditional analogue learning, to a new digitally-enabled hybrid university learning environment (HULE) [3]. Both students and academic staff experience benefits from a digitally-enabled style of learning, as it brings a more up-to-date and less rigid approach to the learning environment [2]. It offers greater and constant access to knowledge and communication streams, with increased flexibility and freedom for students [3]. These benefits have encouraged the widespread mass adoption of hybrid learning at universities [4]. However, the HULE is poorly defined or understood across the literature [5,6], and the term 'hybrid' is used interchangeably with terms such as blended learning, e-learning, or online learning [7,8]. The variations in the language of hybrid learning bring different and mixed understandings both in the literature and in practice. This is largely due to the pandemic accelerating change so rapidly that research has struggled to keep pace.

The COVID pandemic has pushed universities into the hybrid world of education with little time to understand its impacts. Some impacts of this hybrid style of learning on the student experience have only become clear as the pandemic has evolved, since such experiences are difficult to predict or plan for [3,5,9]. There is a particular concern coming from industry research regarding a negative impact on student social connectedness as part of the overall student experience (for example, [1–3,5,7,9]). This is particularly concerning since social connectedness is shown to have a range of positive outcomes for students, including increasing academic achievement and student engagement [10,11]. Further, a lack of social connectedness has negative outcomes for students, including feelings of loneliness or depression, which can reduce academic performance [12,13]. With little time to analyse how student social connectedness is altered in a more digital style of learning at university, a gap has developed in the existing literature. In practice, this has led academic staff and students to be experimenting or improvising as they operate in emergency response, increasing stress, fatigue, and other associated health problems [5,14]. Consequently, greater research is needed to aid academic staff and students in negotiating a new hybrid way of learning.

This article provides a narrative literature review to understand how placemaking theories could help to close gaps in the literature on student social connectedness and the HULE. It applies placemaking theories to bring order, and to develop a framework for understanding student social connectedness in the HULE. Placemaking is applied as both a process and a way of thinking, using urban design principles for improving the quality of places. From a placemaking perspective, the review captures the concept of 'sense of place' to help understand how spaces transition towards places as they become meaningful to people [2]. This focuses on the individual's lived experience, which is a core aspect of placemaking and crucial in understanding student experiences with social connectedness in the HULE [2]. Although placemaking typically concentrates on public places, it has been applied in other places effectively, such as private spaces (for example, see [15,16], and will be applied in this review to the HULE [17]. By doing so, this addresses the main research question of the review: "*How could placemaking structure student social connectedness in hybrid university learning environments?*", which in turn responds to claims surrounding issues of reduced student social connectedness in the HULE. By making an interdisciplinary link between pedagogic concerns, the way of thinking from within social geography, and the theoretical framing from within architecture, this research additionally aims to help educational institutions, design practitioners, and other stakeholders (including funding bodies) better understand how changes to the learning environment might be affecting student social connectedness in universities today and into the future. The review is not meant to be exhaustive, but to provide an insight into the quality and quantity of evidence for student social connectedness in the HULE.

### 1.2. The Evolving Terms of Hybrid Learning and Social Connectedness
#### 1.2.1. Hybrid Learning

There are different understandings of the term 'hybrid learning'. Literature on the HULE is explored differently across and within academic and non-academic research [1,6]. Earlier research by Moore et al. [18] implies that a variety of understandings makes it difficult to understand the type of learning that is expected from a HULE. To avoid confusion and help address the main purpose of this research project, this project adopts the term 'hybrid' as a way of describing all digitally-enabled learning environments that blend both physical and digital learning through interconnection and co-dependence synchronously and asynchronously [2,5,6,19]. Although the terms hybrid or blended are commonly associated with the above description, hybrid is being used here for the following reasons.

- Hybrid learning is viewed situationally rather than as a teaching methodology [20], which takes hybrid learning away from simply the act of learning but enables it to be considered in relation to the environment and its people [21]. This understanding extends beyond a pedagogic lens and enables different forms of learning activities to be present [20]. This makes it typically more recognised beyond pedagogy, with terms like 'hybrid working' from other disciplines likely helping to increase people's familiarity with the term.
- The inclusiveness offered by the term hybrid is often favoured by industry research. Deloitte suggests the term hybrid transcends the term blended by taking a more inclusive understanding of everything that a university institution might offer, rather than simply toggling between face-to-face and online classroom instruction [1,22]. Further, the term 'hybrid' tends to dominate a wide range of industry research, for example, Deloitte [22], Gensler [23], and Times Higher Education [24], whereas in academic level research (including pedagogy), the term hybrid learning seems limited, with blended learning seeming to be used more frequently [25,26].

For these reasons, the term hybrid has been chosen to fundamentally address and enable the exploration of the main aim of this article: "*How could placemaking structure student social connectedness in hybrid university learning environments?*". By addressing these links, it aims to develop a framework using placemaking theories to bring order to understandings of student social connectedness in the HULE, which thus addresses the link between the HULE and student social connectedness. From this understanding, Table 1 was produced to illustrate the relevant dimensions of the HULE for comparisons of different implementations of the HULE. It categorises the dimensions of the HULE into socialisation, space, and time, which addresses the role of the HULE in impacting the social aspect of learning in different physical and digital spaces and across different temporal patterns.

**Table 1.** Dimensions of a hybrid university learning environment (HULE).

| Socialisation | Space | Time |
|---|---|---|
| One-way socialisation | University campus, e.g., lecture hall, classroom, library | Synchronous |
| Bi-directional socialisation | Home working space | Asynchronous |
| Multi-directional socialisation | 'Third' space, e.g., coffee shop | A mixture of synchronous and asynchronous |
| | Digital space | |

1.2.2. Social Connectedness

Social connectedness is described as a subjective feeling of interpersonal closeness in relation to an individual or a group of people within the social world [27,28], and not necessarily to the quantity of an individual's social network. To feel socially connected means to experience a sense of belonging with others, where identification with others is assumed to be linked to this feeling [27]. Social connectedness in students was found to be of significant importance, following earlier findings by Resnick et al. [11] that recognised family and school connectedness to be a strong protective factor against risky behaviours in students. The study measured high school students in the United States from 134 schools and raised awareness of the importance of understanding social connectedness in the learning environment [11]. Later studies have since found a range of positive outcomes for improving social connectedness, including increased academic achievement and student engagement [10,28]. However, with the introduction of technology, understanding how to maintain or develop social connectedness in the transitionary space in-between is particularly difficult as many interactions are unspoken, tacit, and temporary [29]. This space in-between can be referred to as a liminal space. A liminal space can be described as a waiting space, or process of transition and phase in-between [30], and it poses difficulties

in understanding social connectedness in the spaces between online and in-person interactions [29], which is particularly true for the HULE [30]. The aspect of liminal space will be further addressed in the findings section of this review.

Standardising the attributes of social connectedness for the benefit of research is not straightforward and there are various ways of categorising social connectedness based on the context and aims of this study. For instance, Lee and Robbins [31] use a social connectedness scale that features eight statements based on connectedness, affiliation, and companionship, Hare-Duke et al. [32] indicate towards a Thwarted Belongingness subscale of the Interpersonal Needs Questionnaire in mental health contexts, Bailey et al. [33] use a Social Connectedness Index in economic-based research which is based on the number of friendship links on Facebook that one might have, whilst Frieling et al. [34] categorise social connectedness into 3 main attributes: socialising, social support, and sense of belonging. Evidently, there is a disconnect between and within disciplines on how to standardise social connectedness, particularly with the introduction of technologies which bring new forms of connection. As recognised by Frieling et al. [34], standardisation of social connectedness remains limited in a large-scale context, they aimed to simplify these irregularities by categorising social connectedness without being refined to topic-specific confinements. In this review, we apply the 3 attributes of social connectedness taken from Frieling et al. [34] since it offers an effective means of categorising the experiences of social connectedness for students in the university environment without being refined to topic-specific confinements. This is broken down as follows:

(1)　**Socialising**: this is understood as the mixing socially with others and it is measured typically based on the frequency and mode of social interactions that are made.

(2)　**Social support:** this focuses on emotional, informational, and instrumental support that people use to get by. These are broken down as follows: *Emotional support* indicates the assistance that you get from others in terms of care or compassion, e.g., receiving words of praise, empathy, or pats on the back; *Informational support* indicates the assistance that you get from others to receive messages, facts or knowledge, e.g., as advice or feedback on actions; *Instrumental support* indicates the assistance you get from others to meet tangible needs, e.g., borrowing equipment, receiving medical care, or receiving meal preparations.

(3)　**Sense of belonging:** this is a feeling that typically comes when you have access to networks which make you feel a part of society or a community. This can also be linked to a certain place. It can be thought of as the deficit of loneliness or isolation [34].

Whilst these attributes of social connectedness bring a level of standardisation to this fuzzy concept, they generally overlook the significance of place in influencing feelings of social connectedness. These ideas are supported in geographical disciplines where research on the geography of communication by Jansson and Falkheimer [35] requires that the spatial production of place needs an understanding through communication and mediation, since communication produces spaces and spaces produce communication. A study on real-world gaming in Pokemon GO exhibited these relations by highlighting that a sense of social connectedness was produced as players developed a sense of belonging that was linked to a sense of place [36]. This is significant because it recognises the link between social connectedness and place as integral. By introducing the theory of placemaking, these ideas are developed in the following section.

## 2. Theoretical Underpinnings of the Review

### 2.1. Placemaking

Placemaking is defined as "the process of creating quality places that people want to live, work, play and learn in" [37] (p. 2). It is applied as both a continual process and a way of thinking, to help improve the quality of places [17]. It is popular as a community-driven approach, where experts work with members of the community to promote an individual and community sense of place in a collaborative approach [17,38]. To aid in the placemaking process, Wyckoff [37] developed 4 types of placemaking: standard

placemaking, tactical placemaking, strategic placemaking, and creative placemaking [37]. These bring a specific focus, but the end result remains similar, with the development of a quality place being the key outcome. The 'standard', more traditional, type of placemaking is the type that is applied in this narrative literature review. In this way, placemaking occurs typically in incremental ways over a long period of time, however, it can also be sudden with the introduction of large-scale transformative projects or activities to convert a place [37]. Although the traditional focus of placemaking is to explore public spaces, there is evidence that it can also be applied to other spaces, including private spaces, as demonstrated by Larson [15] and Hesjedal [16]. It is in this direction that this review aims to apply placemaking within the HULE.

In this narrative literature review, the framework adopted by Ellery and Ellery [38] in reference to the Project for Public Spaces (PPS) and its placemaking approach will be advocated because of its link to placemaking using an innovative 3-step process to design (which can be applied to the HULE), involving the phases: (1) inspiration, (2) ideation, and (3) implementation. Although these approaches are based on specific principles for creating community places in public spaces within urban areas, such as creating a cohesive vision, translating visions into plans and programs, and ensuring sustainable implementation [37], there is the potential to translate these ideas into the HULE to help develop a framework. Through this review, the aim is to apply these ideas by locating the vision or 'inspiration' phase of a HULE for social connectedness, the attributes of social connectedness in the HULE, the intangible qualities of social connectedness in the HULE, and the measurable data for social connectedness in the HULE, as outlined in Figure 1.

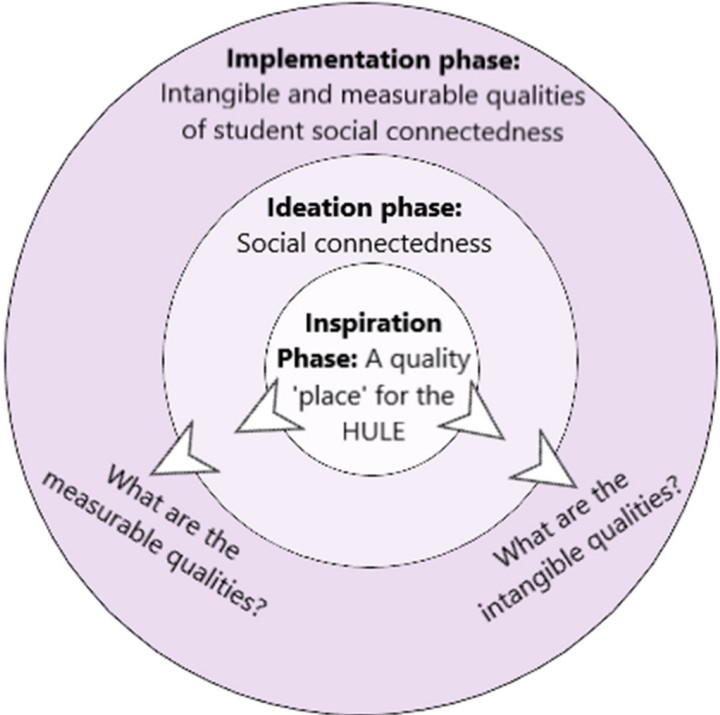

**Figure 1.** Creating a framework based on placemaking theories for developing within the hybrid university learning environment (HULE), inspired by Ellery and Ellery [38].

Building a similar framework based on the guidelines by Ellery and Ellery [38], but orientated towards the HULE, aims to help focus on the challenges of organising the HULE for the student experience in terms of social connectedness. Creating a framework for the HULE is the aim and outcome of this narrative literature review, where Figure 1 becomes further developed in Section 6 based on the findings.

*2.2. Sense of Place*

The key concept of 'sense of place' is applied in the findings section of this review, and is based on a placemaking way of thinking. The concept is centred on a background in social geography to enable the relations between space, place, and people to be explored. Space and place are distinguished in this way through the definitions of Massey [39] and Cresswell [40], with space being defined as a continuous area or expanse of multiplicity where material objects and events occur, and place as something that is produced when humans bring meaning to space, being a specific location, locale, or sense of place. Sense of place is explored in this review through the definition as developed by Swist and Kuwara [2] and Cresswell [40] as follows:

- **Sense of place:** A feeling that typically comes from the human and non-human qualities that fill a space around you. It could include people as a component of the space, or it might simply be the space without people being physically present. The feeling can be positive or negative and continually changing or developing [2,40].

Sense of place is useful to help understand how building the connection between places and individuals do not always bring positive results for everyone, as placemaking can arguably have both positive and negative experiences for individuals [41]. This demonstrates that, whilst the importance of placemaking is often considered a positive process which develops positive perceptions of place [17], it can also be critiqued as a place of destruction [41]. Subsequently, 'sense of place' as a concept within placemaking is useful to apply [38]. Other concepts can also be considered, rather than 'sense of place', within placemaking and will also be indicated in the review to help extend understanding, but 'sense of place' offers a broad range of relations to place, including an identification to self (place identity), an attachment to place (place attachment), a sense of atmosphere (affective atmospheres), and also a feeling of disconnection, placelessness, or more-than-real [42,43]. By integrating a placemaking framework and the overarching concept of 'sense of place', this narrative literature review aims to help unravel the 3 proposed attributes of social connectedness from an innovative perspective.

## 3. The Research Problem and Questions

There is limited academic research on the impacts of the HULE on student social connectedness. The findings from a recent study with architectural industry experts recognised social connectedness as one of the biggest challenges of the HULE [23]. Such a finding is reflected in studies of different spaces, such as workplaces, retail, and care homes, where the balance between digital and physical social connectedness is challenging [44–46]. Research linking social media and social connectedness (or its lack of) has brought questions such as, "Is Facebook making us lonely?" [47], or are people feeling "alone together" [48]. This research highlights the paradox of social media leading to disconnectivity rather than connectivity [49]. The study by Hesselberth [50] considers a 'right to disconnect', which challenges our culture of continuous and constant connectivity, and recognises the importance of enabling opportunities to 'opt out' from connection with social media and 'opt in' with alternative types of connectivity. To some extent, the HULE brings the option for both connectivity and disconnectivity into the classroom by offering digital and physical interaction. Enabling these different forms of connection is a unique factor of the hybrid learning style. Developing further research on socio-spatial understandings of place could widen the opportunity for individual flexibility in learning, and help ensure the social connectedness of students.

Integrating digital and physical interaction in the learning environment is understood by Okita and Schwartz [51] as being especially important, since learning is a social process and requires various forms of communication. As supported by situated learning concepts, learning is situated in a particular social and physical environment [52]. Other research further supports this, suggesting that sharing information within a community helps with the learning process, and that social connectedness positively influences student satisfaction and success rates, and even impacts student health [12,46,53]. Further studies show that

social isolation has significant negative effects on health, with university students being particularly susceptible to feelings of loneliness in their first year [12]. Thus, a high level of importance should be placed on understanding our relationship with technology and social connectedness in the HULE to bring positive student experiences and reduce poor student health or low performance [53]. Yet, most existing studies on hybrid learning and student social connectedness are at the industry level, with little focus on the links between people and place. Further, limited peer-reviewed research is available, which brings an additional problem since peer-reviewed research has long been acclaimed for being devoted to scientific truth and strictly fair, making it a crucial part of the research process [54]. This lack of peer-reviewed research could largely be a result of researchers being yet to agree with the various terminologies or definitions of hybrid learning, making it hard to perform meaningful research [1,18,55]. A lack of knowledge or guidance has led academic staff and students to frequently experiment or improvise as they operate in an emergency response, increasing stress, fatigue, and other associated health problems [5,14].

With our knowledge of the importance of social connectedness in the HULE, it is vital that industry findings surrounding poor social connectedness are addressed and validated by the rigour of peer-reviewed research. Consequently, the key research problems in this review are summarised below:

(1) **Problem 1:** *The attributes of social connectedness lack recognition of place, which is impractical and incomplete.*
(2) **Problem 2:** *Student social connectedness in hybrid university learning environments does not adequately address the connections between the physical space and the digital space.*

To address these research problems, this narrative literature review aims to apply placemaking theories to bring order to the understanding of student social connectedness in the HULE and develop a framework. In this way, the HULE will be explored through the 3 proposed attributes of social connectedness. By applying understandings from placemaking, an innovative way of analysing the HULE is developed. This aims to uncover gaps in the literature on the HULE and propose a more relational way of understanding the HULE and the social connectedness of university students. Providing a framework for the HULE in terms of student social connectedness is the aim and outcome of this narrative literature review. In considering these problems, this review attempts to answer the following research sub-questions:

- **Main RQ:** *How could placemaking structure student social connectedness in hybrid university learning environments?*

This main overarching question will be explored through the following research sub-questions in this article:

(1) **Research Q1:** *How does a 'sense of place' develop the understanding of student social connectedness in the hybrid university learning environment?*
(2) **Research Q2:** *How can social connectedness be understood between physical and digital space?*

## 4. Methods

In this narrative literature review, placemaking is used as a mechanism to explore the context of university students in the HULE with a focus on the outcome of social connectedness. A narrative literature review was chosen because it provides an interpretation and critique of the research in a scholarly summary whilst seeking new understandings that are not yet addressed [54,56]. It has been chosen over a systematic review since it enables the research to be reviewed more theoretically from a broad perspective, and enables the research to be developed with less restraint [54]. Over 400 articles in the fields of pedagogy, social geography and architecture were looked at for this review, with 40 selected articles being included to give an overview of the key findings. This narrow selection process was based on the articles' relevance to social connectedness, placemaking concepts, and the HULE, but articles were specifically chosen for their ability to address the link between

people and place within these topics. Since a placemaking approach is new for examining the HULE, the selection criteria were not straightforward. To compensate for this, the journals examined had to be diverse in covering the key concepts of interest to this review. To do this, the review draws upon the concept of a 'sense of place' from within a placemaking way of thinking to examine the evidence of student social connectedness in the HULE. The key research fields of pedagogy, social geography, and architecture were chosen, since pedagogy is a key part of learning and the environment it takes place in, social geography is key for exploring the social aspects of social connectedness within place, and finally, architecture brings a focus on place and space design.

A search for academic-level empirical studies in the English language was made on Google Scholar and Web of Science using three main keywords: 'student social connectedness', 'hybrid university learning environment', and 'placemaking'. These were combined with deviations from the subsequent keywords: socialising, social support, sense of belonging, and sense of place. The search focuses on the period from 2002 to 2022 to allow for technological developments and changes to the learning environment over the last 20 years. However, emphasis was placed on studies following 2020, to account for the COVID-19 pandemic and the extreme changes that emerged during this time. The inclusion criteria thus requires that studies are (1) reported in English; (2) published in peer-reviewed journals, for the assurance of quality; and (3) published between the years 2002 and 2022. Exclusion criteria included studies not related to the HULE, the university, or student social connectedness. Figure 2 illustrates the process of this search methodology.

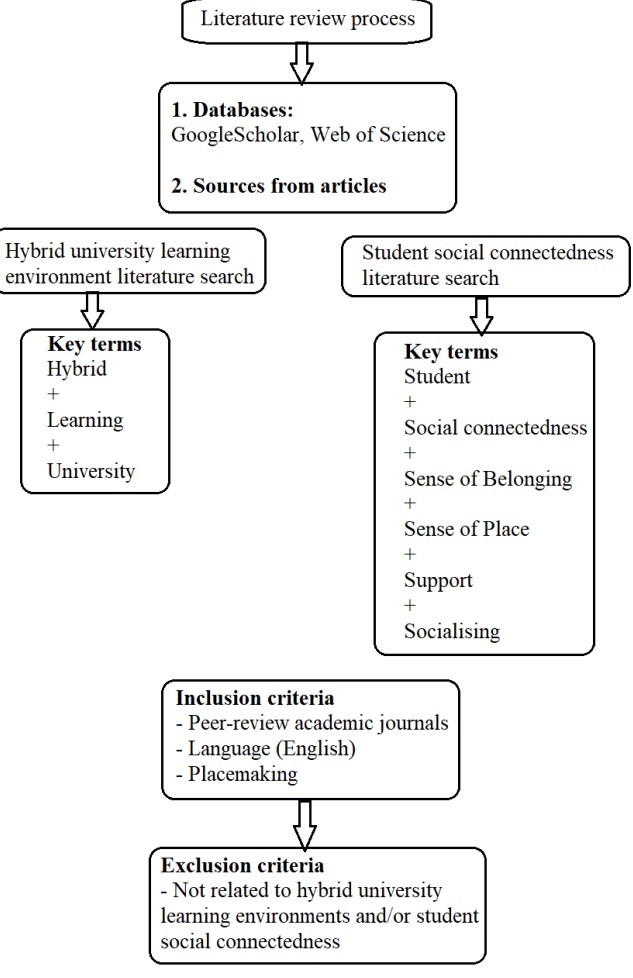

**Figure 2.** Narrative review process.

The findings are presented thematically by dividing the findings section into the 3 proposed attributes of social connectedness: socialising, social support, and a sense of belonging. From these findings, the review synthesises the information by firstly answering research question (1) '*How does a 'sense of place' develop the understanding of student social connectedness in the hybrid university learning environment?*', which addresses the challenge of understanding social connectedness in the absence of place attributes. Secondly, it answers research question (2) '*How can social connectedness be understood between physical and digital space?*', which addresses liminal spaces to highlight the challenge for student social connectedness when linking physical and digital space. In the final section, the main research question is addressed: '*How could placemaking structure student social connectedness in hybrid university learning environments?*', which links understandings of co-design with aspects of placemaking and the framework developed by Ellery and Ellery [38] based on the Project for Public Spaces (PPS) to produce a framework for structuring student social connectedness in the HULE. From synthesising this information, the review finds a lack of clarity in the literature, which highlights a need for more empirical research to answer how student social connectedness is structured in the HULE. The review is not intended to be exhaustive, but it aims to provide insight into the quality and quantity of evidence for social connectedness in the HULE based on the pedagogic, social geographical, and architectural literature.

## 5. Findings: Research Gaps between Placemaking and Social Connectedness

The following section presents the findings from the literature search on placemaking and social connectedness, exploring the 3 proposed attributes: socialising, social support, and sense of belonging. The aim of this section is to outline the key findings, and to explore how placemaking could help organise social connectedness in the HULE by identifying key qualities of the 3 attributes for developing into a framework. Some overlap exists between the attributes, and this is mentioned where possible. In the subsequent section, the results are discussed and synthesised to help answer the research questions in stages: (1) sense of place within social connectedness; (2) liminal space, between in-person and digital space; and (3) structuring student social connectedness through placemaking and co-design. Listed in Table 2 is the included literature from the literature search. The table categorises the literature broadly into pedagogy, social geography, or architecture, which signifies the category of information extracted from the literature for this review. This is based on the literature in line with the research aims of this review. Subsequently, the assigned disciplines are not the intended research discipline of the publication and should not tie the topic to the discipline, but are a representation of how the publication is explored or interpreted in this review under the different disciplines.

**Table 2.** Table of included literature.

| Author(s) and Date of Publication | Pedagogy | Social Geography | Architecture |
|:---:|:---:|:---:|:---:|
| Acton (2017) [21] | X | X | |
| Bilandzic and Johnson (2013) [57] | X | X | |
| Bøjer and Brøns (2022) [20] | X | X | X |
| Bülow (2022) [5] | X | X | X |
| Chayko (2014) [44] | | X | |
| Clarke and Koops (2017) [58] | | X | |
| Ellery and Ellery (2019) [38] | | X | X |
| Ellery et al. (2021) [17] | | X | X |
| Ellison et al. (2007) [53] | | X | |
| Eyal and Gil (2022) [1] | X | | |
| Foth (2017) [59] | | X | X |

**Table 2.** *Cont.*

| Author(s) and Date of Publication | Pedagogy | Social Geography | Architecture |
|---|:---:|:---:|:---:|
| Freberg et al. (2010) [12] | | X | |
| Frick et al. (2020) [29] | | X | |
| Goodyear (2020) [19] | X | X | X |
| Hesjedal (2022) [16] | | X | |
| James and Busher (2013) [30] | X | X | |
| Kohls et al. (2022) [4] | X | | X |
| Kramer (2017) [13] | X | X | X |
| Larson (2021) [15] | | X | X |
| Lee et al. (2011) [28] | | X | |
| Leonard (2014) [42] | | X | |
| Lischer et al. (2020) [60] | X | X | |
| Lupton (2017) [61] | | X | |
| Marta (2019) [62] | X | X | |
| Mäkelä and Leininen (2021) [63] | X | | X |
| McEwan (2011) [49] | X | X | |
| McLean (2020) [43] | | X | |
| Mulcahy et al. (2015) [64] | X | X | |
| Poplin et al. (2017) [65] | | X | |
| Raes et al. (2020) [7] | X | | |
| Rodgers et al. (2020) [66] | | X | X |
| Sandström et al. (2022) [67] | X | | |
| Schwanen and Atkinson (2015) [68] | | X | |
| Singh et al. (2021) [8] | X | X | |
| Skulmowski and Rey (2020) [14] | | X | |
| Swist and Kuswara (2016) [2] | X | X | X |
| Triyason et al. (2020) [9] | X | | |
| Van der Meer et al. (2021) [3] | X | | |
| Westerbeke (2020) [69] | | | X |
| Zydney et al. (2019) [70] | X | | |

### 5.1. Socialising a Sense of Place

Socialising has been examined in placemaking studies that fall outside the remit of the HULE. For example, links between placemaking and socialising are explored in a study of elderly people by Degnen [71] by looking at social memory, embodied knowledge, and the significance of the passage of time. The aim of the study is to help reveal place attachment as a lived reality that is also social. Place attachment is a concept within placemaking, and is outlined as an emotional sense of deep connection with particular places that are experienced by people [71]. In the study, place attachment is found to be not only experienced on an individual level, but as a profoundly social experience. This draws on theories of 'affect' to demonstrate how a place produces a sense of belonging for individuals based on collective social experiences. In the learning environment, Poplin et al. [65] also explore the significant link between place and people through exploring 'power places'. Further, the study by Duff [72] explores how affective atmospheres and placemaking work together to create a place. The link between socialising, place, and 'affect' is also emphasised in the study of interdisciplinary collaboration in research scientists by Hesjedal [16]. Hesjedal [16] applies placemaking to show how socialisation in a place is not only based on geographic location or physical proximity, but social interaction and sensemaking, which is based on certain atmospheric emergences and 'affects', alongside social aspects like body language. This link between people and place is also supported in other research on the learning environment [13,14,62]. Hesjedal [16] states that place, socialisation and the role of affective features and experiences in socialising are key to enabling interdisciplinary research collaboration. Social interaction and sensemaking could be viewed as measurable or intangible qualities for determining the attribute of socialising in the HULE.

By employing the understanding of 'affect' as taken from Degnen [71] and Andrews et al. [73], a place is not just tied to individuals, but it also ties individuals to each other. This way of thinking about the people-place attachment is closely linked with the development of identity, as identified by Eyal and Gil [1], Freberg et al. [12], Larson [15], and Swist and Kuswara [2]. Identity development is understood as being continually in the process of becoming and being reworked or reimagined based on various influences [73]. This could be useful in understanding how experiences with socialising are extended and prolonged in the space between digital and physical spaces through aspects like identity development. By becoming aware of these ongoing social processes that are closely linked with a place, placemaking has the potential to help transform spaces of encounter into places of social interaction and sensemaking [16]. This transformation requires work, and places have to be made by filling them with people, practices, objects, and representations in a reciprocal continual process. Affective atmospheres could further unravel these relations between the individual and the place by exploring how certain emotions emerge and produce various 'affects' [64,73,74]. This is addressed by Lupton [61], to explore how health feels, which conceptualises relations felt between human and non-human actors, and the ways they are perceived in the body in a multisensory approach [61]. Links to the learning environment are made by exploring medical trainees' feelings about practising on virtual patient bodies, but this remains an open question and there is room to expand further on this topic. Addressing these social aspects in spatial processes within a HULE remains largely under-developed [8,14,65], although Acton [21] begins to address the idea through socio-materiality when thinking about a changing learning environment. Socio-materiality is introduced to help expand the notion of the learning environment as a situated process that is entangled and continuously becoming, which needs more development in research on the HULE [21].

*5.2. Social Support Perspectives within Space Diversity*

Social support within placemaking can be explored in the entwined role of social order and power roles in the management system, being formal, informal, or semi-formal [75]. Special facilities are indicated to enable social support, such as conference facilities or enhanced digital accessibility [75]. The study by Bilandzic and Johnson [57] explores the use of digital technology in the library to enhance users' on-site experiences, which recognises the affordances of digital technology as a means of instigating social interaction and support in the physical library space. Di Masso et al. [75] recognise that giving support from 'higher bodies' in a formal setting could trickle down to the users so that they become more self-sufficient with each other to get more informal support between themselves. However, digitalisation has enabled social support to extend beyond the physical place, and it is no longer limited to certain organisational practices in specific time-place orders [9,29,53,62]. Raes et al. [7] indicate that students using remote technology in the HULE can feel excluded, or without social interaction, due to the distance, despite being connected online, which is also supported in the research by Freberg et al. [12], McEwan [49], Skulmowski and Rey [14], and Singh et al. [8]. This feeling of disconnection is shown to increase when technical difficulties occur without immediate support, whereas the students on site might feel neglected by academic staff spending time trying to fix the issue. This issue of reduced social support might also bring additional issues, such as a reduced sense of belonging or social capital [53]. Raes et al. [7] subsequently propose that a technology navigator or operator could be present to assist every class session.

Other means of offering social support with the presence of technology are outlined in the study by Zydney et al. [70], with the development of the project Here or There (HOT). This aims to encourage greater student social connectedness between students online and those in-person, by creating a stronger sense of community. This develops from the possibilities of building for social connectedness physically as afforded by a 'traditional' university (such as large hallways, waiting spaces, and coffee areas) [4,9], but also building for social connectedness digitally using technology as a tool. Further, in a

recent trial project at the University of Twente Library, the redesign of the hallway and coffee corner proposed additional benefits of using digital technology to enable identity development [69]. However, research on technology use as an integrated tool for social interaction remains in its infancy and it is recognised that one of the biggest challenges of technology use in the learning environment is its unpredictable nature [70]. Owing to the complexities of the digital space for social support, it has mainly been explored in digital placemaking literature through co-design [59], and students are found to be structuring the space themselves for their social connectedness needs [5,19]. In this way, the student is enabled to alter their learning environment to best suit their social needs [63,67]. Rodgers et al. [66] and Van der Meer et al. [3] indicate that participation in designing the HULE ensures that users have a level of understanding of new technologies or processes and feel an increased 'sense of community', joint responsibility, and sense of safety within the group. Chayko [44] also recognises the benefits of encouraging identity development in online communities to help develop feelings of safety and escape discrimination. Yet the study by Breek et al. [76] recognises the challenges with two-way communication in online communities, and highlights a need for some level of institutional support. This is supported by the study of Lischer et al. [60], with faculty training being advocated to help transition towards a hybrid style of learning. Further, Bøjer and Brøns [20] recognise that newly built spaces for the purpose of hybrid learning are not always used as planned by the designer, owing to a lack of support or guidance on the space. Yet, they also recognise the potential of co-design as a means of developing the hybrid learning space where the physical space can become part of a 'teacher's pedagogical toolbox' [20]. Thus, careful guidance in co-design seems to be a key quality for determining experiences with social support in the HULE [5,38].

*5.3. Sense of Belonging within Meanings of Place*

The relationship between place, and a need to belong, is outlined effectively in an earlier study of Inuit communities by Leonard [42]. The research is based on the Inuit, an indigenous Arctic community based in Northwest Greenland, and explores their sense of belonging through individuals' sense of social inclusion, interpersonal attachment, and relations with the natural environment through a 'sense of experience and phenomenology of locality' [42] (p. 138 from [77]). These concepts could be viewed as attributes to help measure a sense of belonging in the university student. The landscape in these communities is such that intensive community interaction occurs because communities are isolated by their geography. Identities to a certain group are made and reflected in the individual as a way of maintaining social acceptance and avoiding danger, as found in other research (for example, see [28,66]). This deep bond is intrinsically linked to the environment in both its physical and spiritual sense [42,66]. Recognising place as something beyond its physicality is a crucial aspect that this study illuminates through spirituality, which the HULE needs to address, as digital and physical spaces interact and either alter or maintain a student's sense of belonging [13,17]. Interactions can be experienced in different forms, and the desire and extent for interactions are deemed a highly personal and changeable feeling, as seen in the studies by Lee et al. [28] and Singh et al. [8]. Further, Hesselberth [50] (p. 1994) explores these ideas through the 'right to disconnect'. This both recognises and enables different forms of connections to be made for the benefit of an individual's sense of belonging.

To further explore this sense of belonging beyond physical attributes, Degnen [71] indicates towards the concept of 'insideness', which is broken down into 3 main elements: physical insideness (feeling of attachment to a place), social insideness (sense of belonging or connection to others), and autobiographical insideness (how people narrate their connection to place). A sense of insideness is closely linked to a sense of belonging, but it focuses more closely on distinguishing interactions between the physical, social, and autobiographical. These interactions are indicated to foster attachment to place, and also endorse the continuity of identity [17,71]. As Hesjedal [16] recognises, interactions in place

do not need to be permanent or happen daily, but regular encounters are said to be important in promoting epistemic integration, which indicates the role of time in the HULE [64]. It is suggested that in-person meetings are used as a way of creating a sense of belonging, engagement, community feeling, and trust, which is considered difficult with digital platforms [16]. For instance, digital encounters give no opportunity for 'small talk' by the coffee machine, making it difficult to build personal connections and relationships with new acquaintances. These liminal 'waiting' spaces are recognised by James and Busher [30] as difficult to negotiate in the HULE, a view further supported by Frick et al. [29]. Yet, as indicated in the Inuit study by Leonard [42], placemaking can involve more than just a shared physical place and facilitating face-to-face encounters, it also involves motivating collaboration and developing shared identities which can be done outside of the shared physical place [1,38]. Research from digital geographies on the more-than-real can help to expand these ways of thinking about the digital space as not subordinate to 'real' physical space, but as shaped by key emotive and 'affectual' influences in human-technology relations [42,43]. Clarke and Koops [58] recognise the digital space as a (non-) place to indicate its possibilities of creating a sense of place, despite the physical absence of place, since numerous activities can still occur. This could also be applied to hybrid spaces, as something that is felt and embodied in the process of becoming, yet remains intangible in the spaces in-between [29,59]. In this way, understanding a sense of belonging in the digital and physical space can help to understand how the HULE might be experienced or felt differently by different students.

## 6. Structuring Student Social Connectedness in the Hybrid University Learning Environment (HULE)

In developing these understandings of social connectedness and placemaking theories in the HULE, the following section adopts a specific focus based on the research problems: (1) sense of place within social connectedness; (2) liminal space, between in-person and digital space; and (3) placemaking and co-design. This aims to develop the literature on student social connectedness in the HULE by including the digital space as something beyond simply digital placemaking, but as a hybrid form of placemaking that is an integrated and co-created process felt differently by different individuals. Finally, the main research question of this narrative literature review is addressed: '*How could placemaking structure the social connectedness of students in hybrid university learning environments?*'. A framework is developed grounded on the framework from Ellery and Ellery [38] and based on the Project for Public Spaces (PPS). This links with aspects of co-design to address social connectedness in the HULE from within the proposed 3-step process of design involving the phases: (1) inspiration, (2) ideation, and (3) implementation. This in turn aims to address issues on a bigger scale and respond to industry claims surrounding reduced student social connectedness in the HULE.

### 6.1. Sense of Place within Social Connectedness

This section addresses the role of a 'sense of place' within social connectedness. It aims to highlight the incompleteness of understanding social connectedness in the absence of a physical place. This responds to the initial research sub-question:

- *Research Q1: How does a 'sense of place' develop the understanding of student social connectedness in the hybrid university learning environment?*

In the literature, a 'sense of place' is found to be related to the attributes of social connectedness, with Duff [72] making a direct link with a 'sense of belonging'. A 'sense of belonging' is linked to physical, social, and autobiographical elements, which include place as contingent when developing a place to belong to [71]. In thinking of the body as the site for lived experience, the ways it receives or gives meaning are dependent on its position in space [14,66]. For instance, individual behaviours are regarded as both shaped and being shaped by the space, along with influencing and being influenced by others in the space, which affects the social dynamic and assists in creating a sense of place [2]. In the learning

environment, Swist and Kuswara [2] explore a 'sense of place' in their study of the 'new geographies of learning', to explore the different depths, patterns, and modes of learning engagement in relation to place through a holistic lens. This explores both the online and offline spaces (or "digital and non-digital architecture"), and implies that the ways they are aligned, articulated, and responded to have an influence on the learning environment's 'sense of place' and the creation of belonging. This is further expanded on in the HULE in the study by Poplin et al. [65] on power places ("*places in which people recharge and feel at peace or exuberance, places that evoke positive feelings*" in [65] (p. 76)). The research helps to understand how students feel and describe certain places by mapping the emotions of students in certain places. This links place with emotions but tends to neglect negative experiences with a place, which could be explored through a 'sense of place' [17]. Sense of place enables both positive and negative connotations with a space and opens the door to understanding how a space might negatively impact student emotions [28]. This could add to the understanding of the social connectedness of students in the HULE.

Earlier research by Lee and Robbins [31] indicates that a lack of connection with *society* (a shared group of individuals in the same spatial or social territory) can have an impact on feelings of belonging. This extends the influences of social connectedness to include place in both its physical and abstract form (e.g., a digital place, which can be material, symbolic, or imaginary), further supported by Kramer [13], McEwan [49] and Singh et al. [8]. Poplin et al. [65] also recognises the importance of socialising in place, referring to the campus as a cultural landscape. This highlights the role of creating social spaces to facilitate social processes and develop a place for socialising. In the digital space, the interactions are altered and, as indicated by Freberg et al. [12], Marta [62], Raes et al. [7], Singh et al. [8], and Skulmowski and Rey [14], students might feel excluded, or lacking social interaction or support whilst using remote technology, despite being connected online. The role of a sense of place as both a physical and virtual feeling is therefore significant in maintaining a feeling of support when online [8,9,70]. Thus, it can be seen in the research that numerous beliefs, behaviours and attitudes of individuals interplay within a physical and virtual space to co-constitute a sense of place, albeit it might be felt differently by individuals [28], as indicated through affective atmospheres. This means that, although a 'sense of place' can be influenced and formed by a multitude of people-place encounters, a 'sense of place' remains a highly personal experience in the way that it is felt by the individual. These interactions in place are identified by Swist and Kuswara [2] as having an influence on the development of a sense of belonging, which directs space away from remaining abstract to become something that individuals embody, dwell in, and co-create [13,15]. Thus, in thinking of social connectedness as including a 'sense of place' as a concept, existing understandings of social connectedness can extend to include the integral and unique aspect of place in both its physical and virtual sense [17,21,70]. An understanding of this is important to address how a feeling of social connectedness might develop in the HULE, including in the liminal in-between spaces.

*6.2. Liminal Space: Between In-Person and Digital Space*

This section aims to address the problem of understanding social connectedness in liminal spaces, responding to the literature search that found liminal space in the HULE tricky to apply across in-person and digital spaces. In thinking about how social connections operates in these spaces, this section addresses the sub-research question:

- *Research Q2: How can social connectedness be understood between physical and digital space?*

Designing for social connectedness in the liminal space of a university learning environment requires a shift in understanding when thinking about the HULE. In a 'traditional' university learning environment, building for student social connectedness in liminal space was previously accounted for to some extent physically by offering in-between spaces (such as large hallways, waiting spaces, and coffee areas) for students to communicate, where they would often form crucial network ties [4]. These spaces are important communication nodes which are hard to integrate into the digital space [29], which directly impacts the HULE. The project at the University of Twente library in 2020 shows how a redesign for the hallway and coffee corner might both improve the well-being of users and attract users, with digital technology offering a means of developing identity as a 'bonus' [69]. Although the re-design recognises the importance of liminal spaces in the library, integrating online and in-person places for social connectedness (like identity development) should be seen as an imperative, not as a bonus. Thus, whilst this study has the potential to offer an important contribution towards liminal space in the university learning environment, it fails to account for technology as integral, and transfers back to more 'traditional' ways of thinking about the university learning environment. This demonstrates how integrating social connectedness in the liminal space is less straightforward in the HULE.

Expanding the role of placemaking beyond the physical environment is encouraged in the research by Hesjedal [16] by looking at the facilitation of affective relations among interdisciplinary research scientists. It offers an insightful perspective on the role of affectivity in placemaking and expands placemaking beyond geographical proximity, but the research does not help to understand how to apply or accommodate the digital aspects of the HULE. In online placemaking, the study by Breek et al. [76] recognises the challenges with two-way communication in online communities, but they offer little insight into how to unravel this. Swist and Kuswara [2] also recognise the challenge of arranging personal, social, and material affordances to accommodate the digital aspect of the HULE, and there appear to be enormous benefits of applying 'affective' frameworks to the learning environment, as it opens insight to include spatial aspects with experience [16,68]. The research by Clarke and Koops [58] reveals a digital space to not be something that has physical dimensions (as perhaps virtual reality technologies might give), but instead, it is used metaphorically. This ability to look beyond the physical aspect of a place is further supported by Acton [21] through socio-materiality, which encompasses aspects of a place such as feelings, beliefs, and technologies. Despite the physical absence of place, numerous activities can still occur (including building social connections) and help contribute to the building of a sense of place [53]. In this way, digital space is remote in space and can also be remote in time. It is deemed a 'non-place', which makes it challenging to understand through placemaking concepts [58]. Yet, whilst placemaking typically focuses on the physical place as a geographic location, it is arguably still capable of exploring these 'non-places'. As identified by Hesjedal [16], it is important to look beyond simply physical buildings or architectural structures. This has been enabled more readily as digitalisation has grown. Foth [59] identifies that the opportunities afforded by technology in the mid-1990s led to technology being considered more openly in studies of placemaking, which opens the concept of placemaking to include the digital place and offers a more complete understanding of online and in-between or 'non-place' liminal spaces.

Looking towards concepts from digital geographies of more-than-real places could help to expand the understanding of social connectedness in this liminal in-between space. This adds to the earlier research by James and Busher [30], which recognises

the struggles experienced by a hybrid learning community in navigating the liminal in-between space. Further, the early research from 2013 by Bilandzic and Johnson [57] explores hybrid placemaking in the library as a way of using digital technology to enhance users' on-site experiences. This research, nearly a decade ago, highlights the affordances of digital technology as a tool for placemaking in the physical library space. Although not a recent study of a HULE, this previous strategy of linking digital and physical spaces could offer an impactful contribution by encouraging the embodiment of technology in the library space. Beyond simply adopting technologies to provide access to virtual space (e.g., computers or WiFi), it explores the embodiment of locative media and ubiquitous computing technology in the physical library space as a way of enabling encounters, collaboration, and connected learning among library users. This takes the library away from simply being a place to archive information and knowledge, but as a place of belonging where one can conduct sustained and uninterrupted intellectual work and experience a sense of creativity, inspiration, and scholarship. As a result, Bilandzic and Johnson [57] recognised that libraries are changing to accommodate more social spaces (e.g., couches, lounge areas, and food bars). This study links online and in-person interactions with place to be more than a physical encounter. Thus, students' feelings of social connectedness in place can be developed as something physically tangible and, also imagined, irreal, and occurring in a 'non-place', which incorporates notions of time [49,53,64]. Looking at theories from digital geographies and the more-than-real, helps to understand how affective and emotional forces continue to aid the co-production of 'non-places' when physical proximity is removed. This takes the 'non-places' of digital or in-between space away from being something that is 'unreal' and subordinate to 'real' places, but as something that is an alternative form of place within the 'spiritual' realm [42,43]. In thinking through this lens, our understanding of social connectedness in the HULE could be explored through numerous other crucial nodes of social interaction.

*6.3. Placemaking and Co-Design*

In this final section, the findings and discussion of this narrative literature review are summarised to address the main research question:

- **Main RQ:** *How could placemaking structure student social connectedness in hybrid university learning environments?*

From the research, it emerges that students are actively playing a role in adapting their hybrid learning space to match their requirements, which offers a means for students to structure aspects like social connectedness individually [19]. It is acknowledged by Goodyear [19] that the complexity of design challenges for learning environments is not being addressed adequately through normative models, with its fast-paced and innovative practice preceding the theory. In the HULE, it seems as though spaces are being designed in ways that go beyond the capacities of existing learning design models. An incapacity of current design approaches is likely to have caused academic staff to be experimenting as they go [20,63]. This shift towards co-design was explored in the study by Swist and Kuswara [2] as a way of directing space away from something that is abstract and meaning-less towards something that is embodied, dwelled in, and co-created as a development of an individual's sense of belonging. Bøjer and Brøns [20] explore the potential of co-design as a means of developing the hybrid learning space where the physical space can become part of a 'teacher's pedagogical toolbox'. The research suggests that designing the HULE as part of a participatory design process can help academic staff and students become more environmentally aware and competent in their use of the space. Further, students might become more empowered to use and alter the space accordingly to fit their needs with online and face-to-face learning activities [20,62]. Encouraging strong individual ties to a space is further supported in the literature by Degnen [71], Hesjedal [16], and Larson [15] to bring a greater sense of belonging and sense of place, thus likely impacting social connectedness. This is not an easy task, since personalising the learning experience

adds greater complexity, which is more challenging for universities to manage, and largely explains why it has been so far conducted experimentally [2,62,63].

In recent pedagogic research, Van der Meer et al. [3] produce a guide for integrating social connectedness into online and blended learning communities. The guide outlines design principles to assist lecturers in their role as facilitators and supports a 'willingness to participate' design principle. It is indicated that participation in designing the HULE ensures that users have a level of understanding of new technologies or processes, which could increase feelings of inclusion and connection [15]. This is explored through 'social binding' by Van der Meer et al. [3], as an increase in a 'sense of community', joint responsibility, and sense of safety within the group. Chayko [44] also links social integration with feelings of safety in online communities. Although explored in different contexts, the study on Inuit communities by Leonard [42] also indicates the importance of community in maintaining social acceptance, avoiding danger, and thus experiencing a positive sense of place. As explained above in the more-than-real places, underlying connections constructed in the linking space between online and in-person communication could be made as a way of being both here and there [30]. Adding to the literature on more-than-real places, this ability to develop a sense of place in the spiritual, imagined, and irreal 'non-place' is highly personal and based on the individual [42]. It becomes entangled with both intensely local and global influences, and the communities that form are likely to be continually shifting and imagined differently by each individual [1,2]. In a co-designed HULE, students and academic staff are enabled to shape the space toward their own specific needs. As Bülow [5] recognises, this depends on the knowledge and guidance provided to students and academic staff to make effective co-designing decisions, which has currently led co-design to emerge in the HULE as experimental, and improvised whilst students and academic staff work [19,60,63,67]. Therefore, it seems that co-design in the HULE is not necessarily ineffective, but so far it has been ineffectively executed. Co-design is therefore understood from this review to be a useful solution for accounting for the complexities of the HULE when undertaken with the correct tools and guidance [5].

Placemaking can be used to improve the incorporation of co-design as a participatory approach, with Ellery and Ellery [38] highlighting the use of co-production in the placemaking process for strengthening a community. Ellery and Ellery [38] reference the PPS and its placemaking approach, which adopts a 3-step process to design, involving the phases: (1) inspiration, (2) ideation, and (3) implementation. In blending these ideas of placemaking with co-design for the purpose of the HULE and student social connectedness, the following framework was developed, which is outlined in Figure 3. This outlines the 'inspiration' phase in the centre, with 'place' being used to represent a quality HULE. It is important that this is considered in the in-person, digital, and more-than-real places, as outlined in this review. The 3 attributes of social connectedness are presented around the centre circle as part of the 'ideation' phase, with the following two sections providing the 'implementation' phase, with the inner ring showing intangible measures of social connectedness, and the outer ring showing measurable qualities of the 3 attributes of social connectedness, as based on the findings in this review. In applying this framework in practice, Ellery and Ellery [38] recognise that the initial 'inspiration' phase requires designers to interact with the community to gather relevant information. This first step will be applied in the HULE by undertaking empirical data collection. This will bring a first-hand understanding of social connectedness in the HULE that is not yet available or widespread in peer-reviewed research. Considering the challenges with gathering this information from the literature, it is clear that more empirical research is needed to develop this framework, specifically to answer the question of how student social connectedness is being altered in the HULE as part of the 'measurable qualities' phase. Answering this question requires empirical research and is out of the scope of this literature review. In testing this framework empirically, it is hoped that further understandings can be developed surrounding this research question. Subsequently, Figure 3 acts as a working framework to be added to and adjusted following empirical data collection.

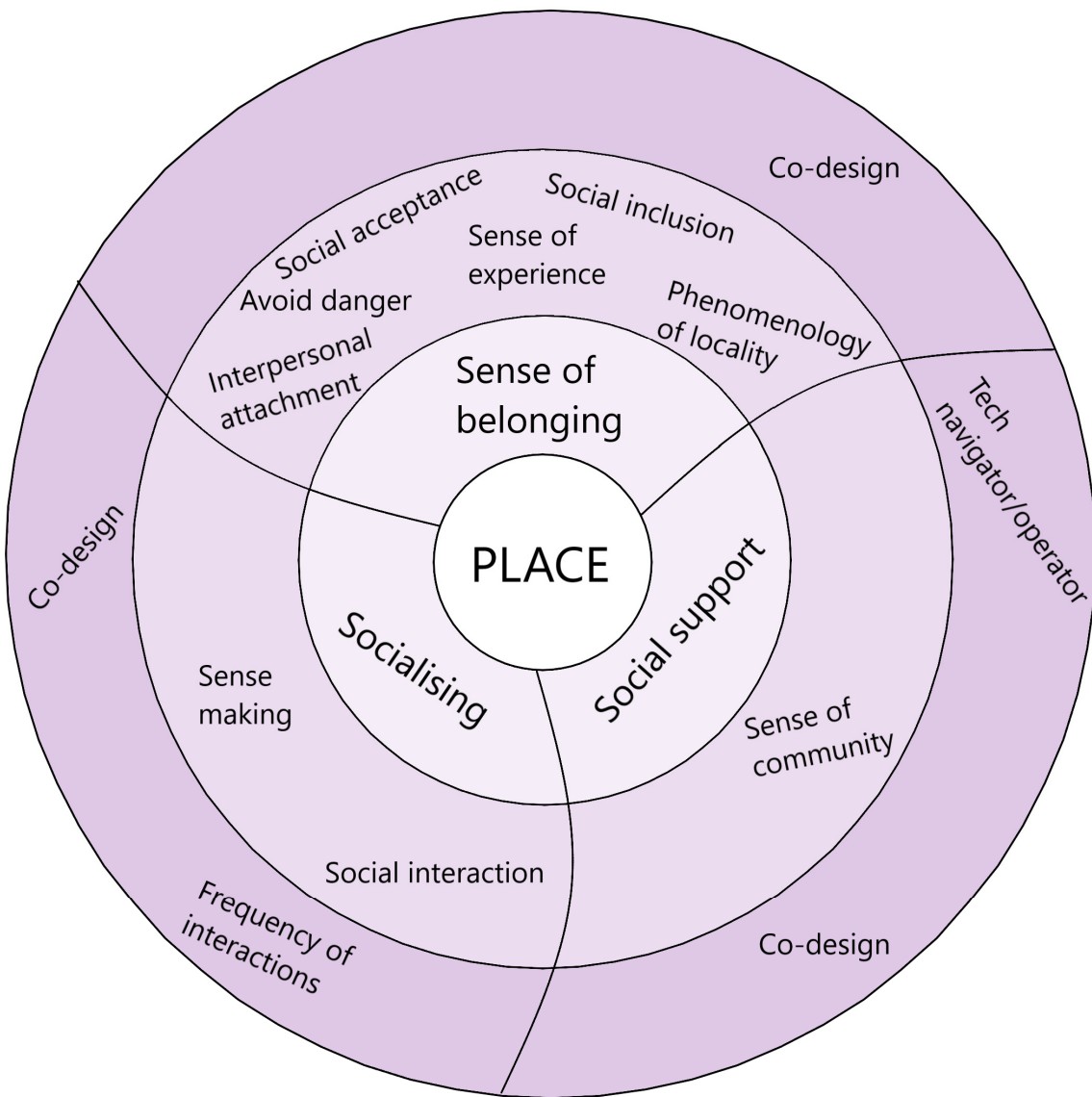

**Figure 3.** A working framework for developing a quality hybrid university learning environment (HULE) for student social connectedness.

## 7. Conclusions

As the first article within a larger doctorate research project on student social connectedness in the HULE, this narrative literature review seeks to answer the main research question:

- *How could placemaking structure the social connectedness of students in hybrid university learning environments?*

Placemaking theories are applied in this review to try to bring order to the topic of student social connectedness in the HULE. This responds to industry claims surrounding a reduced level of student social connectedness in the HULE. The literature search explores peer-reviewed empirical research from the disciplines of pedagogy, social geography, and architecture on the 3 proposed attributes of social connectedness, as developed by Frieling et al. [34]: socialising, social support, and sense of belonging. The review applies the concept of 'sense of place' from within placemaking to reaffirm the importance of place. This responds to the following research problem:

- ***Problem 1:*** *The attributes of social connectedness lack recognition of place, which is impractical and incomplete.*

Whilst this helps address student social connectedness in physical and digital places, understanding how this might function across the physical and digital space is challenging with the introduction of technology in the learning environment. Literature on the liminal space in the HULE addresses these challenges by looking at the spaces in-between. The review then considers social connectedness in the liminal space to respond to the second research problem:

- ***Problem 2:*** *Student social connectedness in hybrid university learning environments does not adequately address the connections between the physical space and the digital space.*

By exploring the attributes of social connectedness from within the physical, digital, and 'spiritual' 'more-than-real' places, this recognises student social connectedness as both a contributing outcome and cause of a place, as part of a lived experience. The challenges of standardising these complex human experiences are addressed through co-design as a way of offering hybrid university students the flexibility that they require. Yet co-design in the learning environment has so far been conducted without the necessary guidance, and placemaking is argued to be a useful contribution to ensuring co-design is conducted effectively. Building on these findings, a framework is built that links social connectedness to the HULE. It offers proposed qualities of the 3 attributes based on the literature, but a lack of clarity in the literature makes it challenging to build a robust framework. Consequently, there is a need for more empirical testing to develop the robustness of the framework and more vigorously address student social connectedness in the HULE. When exploring the findings in practice, theoretical and practical implications could emerge since placemaking frameworks are not well-developed for understanding the spaces 'in-between' the physical and digital space. It is expected that further studies should be reflexive, and could benefit from engaging other theories alongside placemaking, such as those from science and technology studies of socio-materiality, which address the entanglements of social and material aspects to include digital space. Not only could this enhance the ability to study and analyse student social connectedness in the HULE, but it could also help widen the community of those interested in this field of study.

Overall, this narrative literature review offers a preliminary overview of the topic from a new and innovative perspective. Its flexible and explorative approach to analysing the literature aims to expand ways of thinking, which will be followed by empirical testing and a more focused review to explore further the key findings of this review. Whilst the flexible nature of this narrative review helps it to be inclusive and wide-ranging, this approach also acts as a limitation by providing a less systematically rigorous research design. Further, its specific focus on student social connectedness largely overlooks other aspects that might also impact the learning process, such as the impacts of the HULE on academic staffs' feelings of social connectedness, or students' experiences with other challenges in the HULE, such as technology related issues or incompetences [5,8]. Whilst these concerns are not focused on in this review to create greater focus when addressing the concern of reduced student social connectedness, it could also be fruitful for future studies to explore these topics further. Therefore, it should be clear that this review does not claim to be exhaustive, but it is hoped that the impact of this research will expand awareness of the topic and encourage further research. In addition, it is hoped that it will inspire technology developers to continue to create new and exciting technologies that support university student social connectedness both now and in the future.

**Author Contributions:** Conceptualization, T.W., C.L., C.W., L.W. and T.H.; methodology, T.W., C.W. and C.L.; validation, T.W., C.L., C.W., L.W. and T.H.; formal analysis, T.W.; investigation, T.W.; resources, T.W., C.L., C.W., L.W. and T.H.; data curation, T.W.; writing—original draft preparation, T.W.; writing—review and editing, T.W., C.L. and C.W.; visualization, T.W.; supervision, T.W., C.L., C.W., L.W. and T.H.; project administration, T.W. and C.L. All authors have read and agreed to the published version of the manuscript.

**Funding:** This research received no external funding.

**Institutional Review Board Statement:** Not applicable.

**Informed Consent Statement:** Not applicable.

**Data Availability Statement:** The data presented in this study are available upon request from the corresponding authors.

**Conflicts of Interest:** The authors declare no conflict of interest.

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
