# Peer review of "A Narrative Literature Review Using Placemaking Theories to Unravel Student Social Connectedness in Hybrid University Learning Environments"

_buildings, doi:10.3390/buildings13020339_

Round 1

Reviewer 1 Report

1The paper is very well written and straight to the point.  Please pay attention to these two minor corrections.

11. Please discuss the theoretical and practical implications of the findings in detail.

   2. State the key limitations and make recommendations for further studies.

Reviewer 2 Report

The review is very comprehensive and worth sharing in published journal paper.

However, you may improve the paper by providing summary in the form of diagram for each major sections. Although it has been mentioned that the paper is narrative in nature, but having intermittent diagrams may help audience to keep engaged with the narration.

Reviewer 3 Report

This work details a literature review in an admirable attempt to develop a theory about Placemaking in university hybrid classes/spaces. This topic should be of great interest to anyone interested in helping students develop a sense of place in what has become "normal" digital learning spaces.

I appreciate the academic orientation and language of the manuscript but found myself wanting to know a bit more about previous strategies that worked (and more importantly did not work). While this might be the topic of a future publication, it would still be nice to have the authors provide the two most positively impactful practices (and or negative ones). That might also help the authors form a group of interested faculty to help gather empirical information for further studies.
